# Cognitive Slowing, Dysfunction in Verbal Working Memory, Divided Attention and Response Inhibition in Post COVID-19 Condition in Young Adults

**DOI:** 10.3390/life15050821

**Published:** 2025-05-21

**Authors:** Johanna Takács, Darina Deák, Beáta Seregély, Akos Koller

**Affiliations:** 1Department of Social Sciences, Faculty of Health Sciences, Semmelweis University, 1088 Budapest, Hungary; 2Department of Morphology and Physiology, Faculty of Health Sciences, Semmelweis University, 1088 Budapest, Hungary; deakdarina@gmail.com (D.D.); akos.koller@gmail.com (A.K.); 3Department of Physiotherapy, Faculty of Health Sciences, Semmelweis University, 1088 Budapest, Hungary; seregely.beata@semmelweis.hu; 4Department of Physiology, New York Medical College, Valhalla, NY 10595, USA; 5Research Center for Sports Physiology, Hungarian University of Sports Science, 1123 Budapest, Hungary; 6Department of Translational Medicine, Faculty of Medicine, HUN-REN-SE Cerebrovascular and Neurocognitive Disease Research Group, Semmelweis University, 1094 Budapest, Hungary

**Keywords:** cognitive dysfunction, post COVID-19 condition, long COVID, verbal working memory, executive function, young adults

## Abstract

After COVID-19 infection, about 30% of people have clinically persisting symptoms, characterized as Post COVID-19 Condition (PCC). One of the most reported symptoms in PCC is cognitive dysfunction, yet there are only a few studies investigating long-term effects on different domains of cognitive function. A total of 107 young adults, university students aged 18–34 years, participated. In total, 68.2% had contracted SARS-CoV-2; 21.9% showed PCC. Three groups were compared: no-C19 (COVID-19-negative controls), C19 (COVID-19-recovered without PCC) and PCC. Attention and executive function were measured with the Vienna Test System (Schuhfried^®^, Mödling, Austria). In verbal working memory, the PCC group had a significantly lower performance with a moderate effect. The rate of below-average performance was higher in PCC (56.2%) compared to no-C19 (20.6%) and C19 (15.8%). In divided attention and response inhibition, PCC also showed lower performance, 62.5% and 37.5%, respectively, than no-C19 and C19. The co-occurrence of decreased cognitive functions was pronounced in PCC. The present study revealed significant long-lasting cognitive dysfunction in PCC in young adults, two years after COVID-19 infection. Verbal working memory was significantly impaired, and a lower performance was found in divided attention and response inhibition. In addition, there was an increased reaction time in most cognitive tasks, demonstrating cognitive slowing in young people with PCC.

## 1. Introduction

Since the outbreak of the SARS-CoV-2 coronavirus in 2020, over 770 million people have been infected by the virus worldwide [1], which causes a multi-organ disease (COVID-19). Following COVID-19 infection, clinically significant ongoing, new or returning symptoms may occur, causing disability and decreased quality of life, a syndrome called Post COVID-19 Condition (PCC). PCC is a heterogeneous disease associated with a wide range of clinical manifestations [2,3,4,5,6,7,8].

PCC is more common after the more severe forms of COVID-19, but it can also be detected after an infection with mild symptoms [9]. PCC is primarily observed in older adults and women but it also affects young people [10,11,12]. There is great variability in the estimated prevalence [13]. Based on previous reports, the prevalence of PCC is around 30% of COVID-19 infected individuals [14,15,16], and about 4% to 16.5% in the young adult population [17,18]. In non-hospitalized patients, the estimates range from 10 to 35%, and it affects asymptomatic cases [8,19,20]. According to previous studies [21,22,23], vaccination, with two doses, reduces the incidence of PCC and its aftereffects even in people who have other risk factors.

Assessments of Post COVID-19 symptoms are based on mostly self-reported questionnaires and vary from patient to patient, which makes diagnosis difficult. Accordingly, the most common complaints in PCC are neurological and psychiatric symptoms, such as fatigue, headache, dyspnea, anosmia, cognitive impairment, concentration and memory problems, sleep distribution, dizziness/vertigo/balance and problems with mood and emotion, independent of hospitalization [24,25,26,27,28,29,30,31,32]. After mild COVID-19 infection, one of the most reported symptoms is cognitive dysfunction, even in young adults and persons with no underlying medical diseases [20,25,30,32,33,34].

There have been several previous reports regarding the effects of COVID-19 on cognitive functioning, on primarily the older/elderly population, in which the disease had more serious psychosomatic effects, morbidity and mortality [13,14,15,16,17,18,19,20]. Because of the somewhat subtle effects of COVID-19 on young people, it is easy to miss PCC in this population; however, the long-term effects may hinder their educational advancement. Thus, it is important to reveal and recognize the potential development of PCC in this valuable young population, i.e., university students, in an objective manner.

In addition, increasing evidence suggests that cognitive dysfunction also persists after COVID-19 infection for more than three months and can last longer than other symptoms. However, only a few studies have systematically examined long-term cognitive impairments after COVID-19 infection [35,36,37] and focused on the different domains of cognitive functions in PCC [38,39]; these studies found impairment in attention and executive function. There is one cognitive impairment, slow processing speed, in PCC that has received some interest recently. Numerous studies in both the acute stages of COVID-19 and PCC have documented this, particularly in those who self-report cognitive impairments [35,36,37,38,39]. At the same time, in most of the cases, the examined sample included a high rate of hospitalized people with moderate/severe COVID-19, making it difficult to draw conclusions on non-hospitalized people and a younger population. Furthermore, most studies in this field examine adults/older adults with significant previous comorbidities; thus, it is difficult to distinguish whether the cognitive impairment found corresponds to a mild impairment or dementia, or is the consequence of other diseases [30,35,36,37,40].

Based on the above, we hypothesized that long-term cognitive dysfunction exists in young adults with PCC compared to those who were not infected and who had COVID-19 infection but did not experience PCC after recovery. Thus, in the present study, we aimed to assess cognitive functions in young adults with PCC using a PC-based, normed, age- and gender-adjusted cognitive test battery (Vienna Test System Neuro module) measuring different domains of cognitive functions, such as attention and executive function.

## 2. Materials and Methods

### 2.1. Study Sample

To ensure homogeneity of the sample, the participants were recruited from university students, and those who had previously documented medical histories of chronic diseases and neurological and psychiatric conditions in the year before or at the time of the study were excluded. The study sample included 107 young adults, university students, aged between 18 and 34 years (*M* = 21.5 ± 3.3), 79.4% female (n = 85) and 87.9% with secondary education (n = 94) with similar living situations and without comorbidities (previously documented medical histories of chronic diseases and neurological and psychiatric conditions in the year before the study were excluded from the study).

Based on the data from the WHO Post COVID-19 Case Report Form (Post COVID-19 CRF), 68.2% (n = 73) had contracted SARS-CoV-2. In total, 86.3% (n = 63) of those had had SARS-CoV-2 infection confirmed by a diagnostic test: positive PCR test during the acute illness and/or positive antigen test (rapid test) during acute illness and/or positive antibody test during/after the acute illness. The further 10 participants had had symptomatic acute COVID-19 without a diagnostic test and had received a diagnosis of COVID-19 by a healthcare worker during the acute illness. Based on the classification of the WHO in most cases, the infection was mild (89%, n = 65). In total, 21.9% (n = 16) of infected young adults showed persisting, still present/intermittent symptoms.

During analysis, three groups were compared: no-C19—COVID-19-negative controls (34/107, 31.8%); C19—COVID-19-recovered without PCC (57/107, 68.2%); and the Post COVID-19 Condition (PCC, 16/73, 21.9%) groups (see Appendix A).

### 2.2. Measurements

Cognitive assessment was conducted with a PC-based test battery (Vienna Test System Neuro module, Schuhfried^®^, Mödling, Austria) measuring the different domains of cognitive functions: attention (domains: intrinsic alertness, divided attention and processing speed) and executive function (cognitive flexibility, verbal working memory and response inhibition). Test results were interpreted in terms of percentile ranks (PRs), and age- and gender-adjusted norm-referenced scores, which allows the comparison of the test results of different samples. From the percentile rank, it is possible to determine whether a respondent’s score is above average (76–100 PR), average (25–75 PR) or below-average (0–24 PR) compared to the comparison group in question [41]. For detailed information about the cognitive tasks, and the main and subsidiary variables of the measured domains of cognitive functions, see Appendix A.

Cognitive tasks were performed in the morning between 9 am and 1 pm, before the main meal. The settings of the workstation were based on the Vienna Test System manual about “The Workstation—Ergonomics (Schuhfried^®^, Austria)”, including settings for desk and chair, lighting, noise, temperature and break to ensure reproducibility. The desk and the seat height of the chair were adjusted for each participant to sit in an upright position during the tests, with about a 30° view angle of the screen. The workplace was lit by natural and adequate artificial light, i.e., the degree of contrast between the computer screen and the rest of the work environment was appropriate, and the lightning did not cause glare or reflections in the monitor. To exclude distractions, participants completed the cognitive assessment in a separate, spacious, calm and quiet room without air conditioning, where the room temperature was below 25° but not lower than 19°. The instructions of each test were presented in standardized form on-screen, i.e., the test administrator did not need to explain the test items in detail, and the participants completed the cognitive assessment in an unsupervised setting. After the main instructions, how to use the system and the response panel, and giving a general description of the tests, the test administrator left the room until the end of the measurement.

The assessment of the infection and symptoms during (acute COVID-19) and after (Post COVID-19 condition) COVID-19 infection were collected via an online survey using the WHO Post COVID-19 Case Report Form (Post COVID-19 CRF) (https://www.who.int/publications/i/item/global-covid-19-clinical-platform-case-report-form-(crf)-for-post-covid-conditions-(post-covid-19-crf-) (15 March 2022). The Post COVID-19 CRF includes 50 different symptoms. This Case Report Form also assessed a quality of life (functioning, QoL-F) component measuring difficulties for the past seven days on a 5-grade scale from “no difficulty” to “extreme difficulty/cannot do” (lower value better QoL-F), such as standing for long periods, learning new tasks, maintaining a friendship, etc. The online questionnaire was completed after the cognitive assessment.

Study data were collected from 27 July 2022 to 13 June 2023. The assessments were conducted on average 1.4 years (SD = 0.73) after the COVID-19 infection. The data were managed using REDCap electronic data capture tools hosted at Semmelweis University [42,43]. We have been granted Regional, Institutional Scientific and Research Ethics Committee Semmelweis University (SE RKEB) ethical approval (No. SE RKEB 118/2022).

### 2.3. Statistical Analysis

Descriptive statistics are reported as mean, standard deviation, median and relative frequencies. To examine group differences, robust independent-sample t-tests with Hedges’ g effect size measurement and robust one-way ANOVA with partial eta squared effect size measurement were used due to the unequal sample sizes. Effect size measurements are reported with a 95% confidence interval. In robust one-way ANOVA, the Games–Howell post hoc test was used in multiple comparisons for observed means. For testing association, Pearson’s chi-squared test was applied with Cramer’s V as the measure of the strength of association. The level of significance was set a priori at 0.05. A priori power analysis was used to calculate the required total sample size. For one-way omnibus ANOVA with fixed effects, the minimum total sample size = 102, with the following parameters: *f* = 0.40 (calculated based on the effect size from variance using direct partial *η*^2^ = 0.14 (“as in SPSS”), *α* = 0.05, *power* = 0.95 and number of groups = 3. The minimum sample size was respected in the present study but there were unequal sample sizes; however, during the analysis, unequal variances were not observed. Statistical analyses and visualization were conducted using IBM SPSS Statistics for Windows, Version 25.0 (IBM Corp. Released 2017, Armonk, NY, USA), jamovi (Version 2.2.2, The jamovi project 2021) and qgraph package (version 1.6.9.) in R.

## 3. Results

### 3.1. Characteristics of the Study Sample

The study sample included 107 young adults, aged between 18 and 34 years (*M* = 21.5 ± 3.3), 79.4% female (*n* = 85) and 87.9% with secondary education (*n* = 94). During analysis, three groups were compared: no-C19—COVID-19-negative controls (34/107, 31.8%); C19—COVID-19-recovered without PCC (57/107, 68.2%); and the Post COVID-19 Condition (PCC, 16/73, 21.9%) groups. The three groups did not show significant differences in age (*F*(2,104) = 2.301, *p* = 0.114) and education (*χ*^2^(2, N = 107) = 0.731, *p* = 0.694).

The total mean score of quality of life (functioning, QoL-F), which ranged between 1 and 5 (lower value, better QoL-F), revealed a statistically significant difference between the three groups with a small effect (*F*(2,104) = 3.798, *p* = 0.026, *η*^2^*_p_* = 0.07 [0.005; 0.15]). The control group showed a significantly higher value (*M* = 1.45, *SD* = 0.37) than the infected group (*M* = 1.26, *SD* = 0.33) (*p* = 0.030). PCC showed an intermediate value (*M* = 1.41, *SD* = 0.28). Based on the mean values, the three groups had a lower value than 2, which indicated no/mild difficulties, i.e., good QoL-F for the past seven days before the measurements.

### 3.2. Symptoms in Acute COVID-19 (ACC) and Post COVID-19 Condition (PCC)

In ACC, the average number of symptoms per individual was 11.6 (1–48, *SD* = 9.9). The most frequently reported symptoms were neurological and psychological: persistent fatigue and depressed mood, occurring in 50% of the young adults (Figure 1).

In PCC, the average number of persisting, still present/intermittent symptoms per individual was 9.6 (1–28, *SD* = 7.8). The leading complaints were neurological: trouble in concentrating occurring in 68.8% of the young adults. Persistent fatigue and forgetfulness were also commonly reported, with a prevalence of 62.5% and 50% (Figure 1; for the detailed frequency data of symptoms, see Appendix A).

The number of ACC symptoms was higher, with a small effect in the PCC group (*M* = 13.81, *SD* = 8.68) compared to the C19 group (*M* = 10.88, *SD* = 10.19) (*t*(61) = 1.123, *p* = 0.270, *g* = 0.30 [0.27; 0.86]). The number of symptoms in ACC and PCC showed a significant positive large correlation (*r*(12) = 0.798, *p* < 0.001).

### 3.3. Results of the Assessment of Cognitive Functions

The assessments were conducted on average 1.4 years (SD = 0.73) after the COVID-19 infection. The time since infection did not show a significant difference between the C19 (*M* = 1.43, *SD* = 0.73, *min–max* = 0.36–3.20) and PCC groups (*M* = 1.30, *SD* = 0.72, *min–max* = 0.40–3.00) (*t*(71) = −0.593, *p* = 0.555). The duration of the cognitive assessment was about 23 min; participants completed the assessment in 26.3 min on average (*SD* = 1.8). There was a non-significant difference between the no-C19, C19 and PCC groups in the total working time (*F*(2,104) = 1.009, *p* = 0.368, *η*^2^*_p_* = 0.01 [0.00; 0.07]).

#### 3.3.1. Attention

In the case of *divided attention*, the no-C19, C19 and PCC groups showed a statistically non-significant difference in percentile rank, but the PCC group revealed lower performance with a small effect (*F*(2,104) = 1.201, *p* = 0.305, *η*^2^*_p_* = 0.02 [0.005; 0.08]). A higher rate of below-average performance was found in the PCC (62.5%) and in the C19 (57.9%) groups compared to the no-C19 group (41.2%) (*χ*^2^(2, N = 107) = 3.035, *p* = 0.219, *V* = 0.17) (Figure 2), revealing a longer median reaction time (Appendix A). In addition, the time since infection did not show a significant difference between the “below-average” (*M* = 1.33, *SD* = 0.70) and “average/above average” (*M* = 1.52, *SD* = 0.77) performance groups (*t*(71) = 1.052, *p* = 0.298).

*Intrinsic alertness* and *processing speed* showed a non-significant difference between the no-C19, C19 and PCC groups. The rate of below-average scores was about 10% in intrinsic alertness and nearly 50% in processing speed, irrespective of the groups. For the detailed results of attention functions in the main and subsidiary variables, see Appendix A.

#### 3.3.2. Executive Function

The *response inhibition* percentile rank did not show a significant difference between the no-C19, C19 and PCC groups. However, the PCC group showed a higher rate of below-average performance (37.5%) compared to the C19 (19.3%) and no-C19 (14.7%) groups (*χ*^2^(2, N = 107) = 3.580, *p* = 0.167, *V* = 0.18), a more frequently unsuccessful inhibition of no-go stimuli with an increase in the mean reaction time and in the standard deviation of reaction time (Appendix A). Examining the effect of the time since infection, there was a non-significant difference between the “below-average” (*M* = 1.26, *SD* = 0.73) and “average/above average” (*M* = 1.45, *SD* = 0.73) performance groups (*t*(71) = 0.880, *p* = 0.387).

*Verbal working memory* showed a statistically significant difference between the no-C19, C19 and PCC groups (*F*(2,104) = 3.917, *p* = 0.023, *η*^2^*_p_* = 0.07 [0.005; 0.15]). The PCC group had a lower performance with a moderate effect compared to the no-C19 and C19 groups. The rate of below-average performance was significantly higher in the PCC group (56.2%) compared to the no-C19 (20.6%) and C19 (15.8%) groups (*χ*^2^(2, N = 107) = 11.637, *p* = 0.003, *V* = 0.33) (Figure 3). Furthermore, the time since infection did not show a significant difference between the “below-average” (*M* = 1.26, *SD* = 0.73) and “average/above average” (*M* = 1.47, *SD* = 0.73) performance groups (*t*(71) = 1.731, *p* = 0.095).

The lower performance in the PCC group was the result of a significantly higher number of omissions and higher mean time “correct” stimuli compared to the no-C19 and C19 groups (Figure 4A,B). At the same time, the number of false alarms (errors/incorrect responses) and working time showed a non-significant difference between the groups (Appendix A).

A non-significant difference in *cognitive flexibility* was revealed between the no-C19, C19 and PCC groups. The rate of below-average performance was about 20–25% in all groups. For detailed results on the subsidiary variables, see Appendix A.

#### 3.3.3. Co-Occurrence of Decreased Cognitive Functions

Finally, the co-occurrence of decreased cognitive functions, revealing below-average performance in cognitive functions, was examined. Overall, half of the PCC group (50.1%) showed below-average performance, and at least three or more out of the six cognitive functions measured were impaired (no-C19: 20.5%, C19: 22.8%, *χ*^2^(2, N = 107) = 5.585, *p* = 0.061, *V* = 0.23). The co-occurrence of the decreased cognitive functions is depicted in Figure 5. Pronounced co-occurrence was shown in the PCC group. In at least one-third of young adults, the below-average performance of verbal working memory, processing speed, divided attention and response inhibition were associated with each other.

## 4. Discussion

The present study, using PC-based, normed, age- and gender-adjusted cognitive tests, assessed cognitive functions in three groups, no-C19 (COVID-19-negative controls), C19 (COVID-19-recovered without PCC) and PCC (Post COVID-19 Condition) groups, and found that there are problems in several domains of cognitive functioning in PCC among young adults on average 1.4 years after COVID-19 infection. In addition, the co-occurrence of decreased cognitive functions revealed that PCC affects several cognitive functions at the same time. Finally, based on the results, the lower performance in PCC is related to fluctuation in attention, not a relapse of attention, resulting in increased reaction time in most cognitive tasks, which also demonstrates cognitive slowing, i.e., increased time to process information and respond to it, in young people. In the PCC group, the proportion of below-average reaction time was 68.8% in divided attention (no-C19: 47.1%, C19: 56.1), 56.2% in response inhibition (no-C19: 41.2%, C19: 35.1) and 37.5% in verbal working memory (no-C19: 8.8%, C19: 5.3%). In cognitive flexibility, PCC and C19 showed 25% and 24.6% below-average performance (no-C19: 17.6). Thus, the findings of the present study suggest, in line with our hypothesis, that long-term cognitive dysfunction exists in young adults with PCC compared to those who were not infected and who had COVID-19 infection but did not experience PCC after recovery.

Previous studies examining cognitive functioning in PCC found that executive function, memory and attention are mainly affected [37,44,45,46,47,48]. However, regarding the more long-term effects of PCC on cognitive functioning, the results are less consistent [49,50,51]. A recent study found cognitive slowing, increased reaction time, in PCC on average one year after COVID-19 infection, which can be considered a cognitive signature of PCC [52].

The present study revealed that young adults with PCC have significant long-term deficits in cognitive functioning after on average 1.4 years after COVID-19 infection. Specifically, verbal working memory showed significant impairments in PCC, with a higher mean reaction time (in sec) and a higher number of omissions (Figure 3 and Figure 4). Based on previous studies, verbal working memory can be impaired after COVID-19 infection, but the long-lasting impairments are less consistent. It is assumed that working memory deficits can improve over time, although impairments may linger in people who have persistent symptoms [38,49,53,54].

It is well known that verbal working memory is responsible for temporary storage of verbal information, such as letters, words, numbers or nameable objects. Based on modern theories of working memory, it is not only the interface between perception and long-term memory; it is also the central relay in controlling complex cognitions [55,56]. The measurement of verbal working memory with the Vienna Test System—used in the present study—covers the four main executive functions of verbal working memory in addition to storage and rehearsal processes [57,58]: checking and updating the contents of short-term memory, shifting the focus of attention between different tasks and functions, continuous monitoring and coding the contents of working memory about the place and time of storage and inhibiting dominant response tendencies. The deficits in verbal working memory revealed in the present study in the PCC group indicate a possible impairment of these executive functions in young adults.

In line with a previous study that examined cognitive inhibition, and highlighted long-lasting inhibition deficits in PCC [59], we also found a lower performance in response inhibition in the PCC group compared to the C19 and no-C19 groups, with a more frequently unsuccessful inhibition and an increased mean reaction time. In divided attention, lower performance was found, with a longer median reaction time, in young adults after COVID-19 infection in the PCC and C19 groups, where more than half of the participants showed decreased performance in divided attention, compared to the no-C19 group (Figure 2). Young adults with PCC can have difficulties in situations in which several different things are required of them simultaneously. This result is in line with a study by Bertulecci et al. [54], who found that sustained and divided attention are frequently impaired in PCC.

Finally, the present study found that significant deficits in cognitive functions are consistent with the leading complaints in PCC (Figure 1). About 70% of the PCC group reported trouble in concentrating and half of them showed still present/intermittent forgetfulness. Our results are in line with previous studies [40,47,60], which revealed an association between cognitive dysfunction and reported cognitive complaints. An important novel finding of the present study is that this association can be observed even among young adults, university students. A recent study by Bland et al. [61] found that there is no correlation between objectively and subjectively measured cognitive functions among adults. Subjective complaints did not show a difference between the groups of “post COVID-19”, “recovered COVID-19” and “no COVID-19”, taking into account stress and fatigue. In the present study, stress/anxiety were not leading complaints in ACC and PCC but fatigue was more prevalent. However, the study by Bland et al. [61] also demonstrated that objective cognitive deficits could not be explained by fatigue or stress but were significantly related to COVID-19 status. A large portion of the PCC group also showed persistent fatigue, which cannot relate to physical, but rather mental, fatigue/exhaustion, which may not affect other cognitive functions. Indeed, we did not find differences between the groups in intrinsic alertness, processing speed and total working time during cognitive assessment in young adults. Previous studies also did not find a correlation between fatigue and cognitive impairment, showing that fatigue and cognitive dysfunction are the most common but different symptoms of COVID-19 [62,63]. The association between objectively (with neurocognitive tasks and clinical assessment) and subjectively (with self-reports) measured cognitive difficulties following COVID-19 is still in question; they are two distinct measures of cognition. Thus, in addition to self-reported measures, one may consider taking an objective assessment of cognitive functioning in PCC even after mild infection and in a younger population with and without reported cognitive complaints.

Based on our findings, we entertained the idea that certain cognitive dysfunctions are mechanistically related to PCC (Figure 5). Indeed, the co-occurrence of decreased cognitive functions, such as verbal working memory, divided attention and response inhibition, was very pronounced in PCC, suggesting that some neural connections are common and may relate to certain brain regions (prefrontal cortex) [64,65]. Further neurological studies with fMRI should reveal such potential connections, which may provide a mechanistic basis for interventions in PCC [66,67].

In summary, in the present study, significant long-lasting cognitive deficits were found in PCC in young adults two years after COVID-19 infection. Specifically, verbal working memory was significantly impaired and a lower performance was found in divided attention and response inhibition. Based on the results, the lower performance also demonstrates cognitive slowing, increased reaction time, in young people with PCC.

### Limitations

The present study has some limitations. There is a possible selection bias as well as a recall bias since this study was conducted around two years after COVID-19 infection. It is important to highlight that it is difficult to distinguish the symptoms associated with PCC from the symptoms related to the pandemic. In addition, there are no pre-COVID-19 cognitive data about the cognitive functions in the study sample; at the same time, during the interpretation of the cognitive performance, age- and gender-adjusted norm-referenced scores were used, which allows the comparison of the test results of different samples. This study is a cross-sectional study; thus, we did not have any information about the changes in the symptoms of the Post COVID-19 condition and cognitive functioning. An additional limitation is the small sample size and unequal sample sizes; even though robust tests were used during analysis and unequal variances were not observed, this fact restricts the generalizability of the findings. Finally, in the present study, potential confounders, such as psychosocial factors, which are also related to cognitive functions, were not examined. Nevertheless, those with previously documented medical histories of chronic diseases and neurological and psychiatric conditions in the year before this study were excluded from this study. Based on a recently published study, examining which individuals are at risk of developing Post COVID-19 Condition (PCC), female sex, older age, higher body mass index, smoking, pre-existing comorbidities, including anxiety and/or depression, asthma, COPD, diabetes, IHD and immunosuppression, and previous hospitalization or ICU admission were the significantly associated risk factors [23]. Thus, it is important to further investigate the potential confounders that can be related to cognitive functions and the vulnerability of PCC. Nevertheless, despite these limitations, the present study has many strengths: the homogeneous sample of young adults with similar characteristics of COVID-19 infection, without comorbidities and with similar living situations that may exclude several biases of potential confounders.

## 5. Conclusions

The present study revealed that in young adults, university students, without other comorbidities, there can be significant long-lasting cognitive impairments two years after COVID-19 infection. Specifically, verbal working memory was significantly impaired and a lower performance was detected in divided attention and response inhibition. The observed increased reaction time in all cognitive tasks may demonstrate cognitive slowing in young people with PCC. It is important to highlight the fact that there can be other factors and potential confounders that can result in below-average performance in cognitive functions among young adults. Thus, future studies should follow the progression of cognitive impairments, including further factors/confounders, and should develop interventions to improve cognitive functioning in PCC for all ages.

## Figures and Tables

**Figure 1 life-15-00821-f001:**
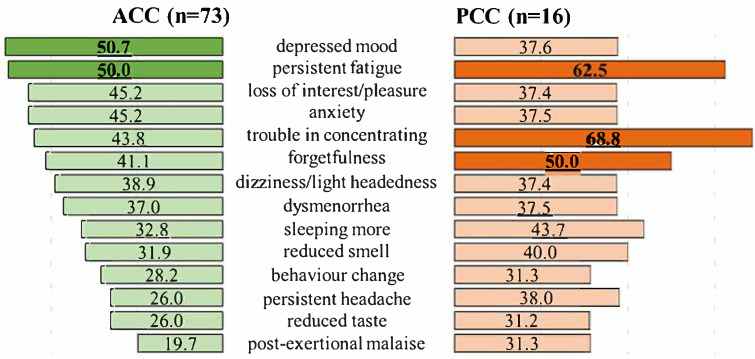
The frequency of various symptoms (%) in acute COVID-19 (ACC, A; C19 + PCC groups) and Post COVID-19 Condition (PCC, B; PCC groups). Symptoms with a frequency of at least 30% in ACC and/or PCC are shown. Frequency of symptoms ≥ 50% are in dark and bold.

**Figure 2 life-15-00821-f002:**
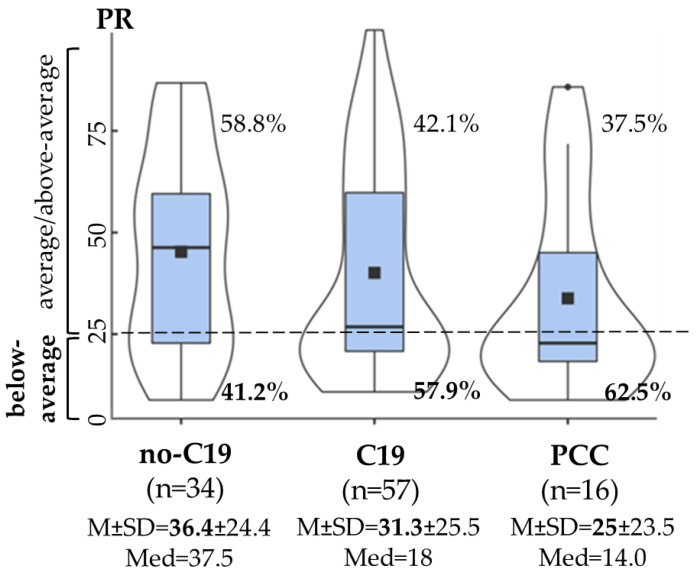
Divided attention in the no-C19 (COVID-19-negative controls), C19 (COVID-19-recovered without PCC) and PCC groups. Violin plots with density curves and boxplots of percentile rank (PR). The horizontal line in the boxplot is the median of the PR; the square is the mean of the PR. M ± SD: mean ± standard deviation, Med: median. Dashed line: PR = 25, below-average performance: PR < 25, average/above-average performance: PR ≥ 25.

**Figure 3 life-15-00821-f003:**
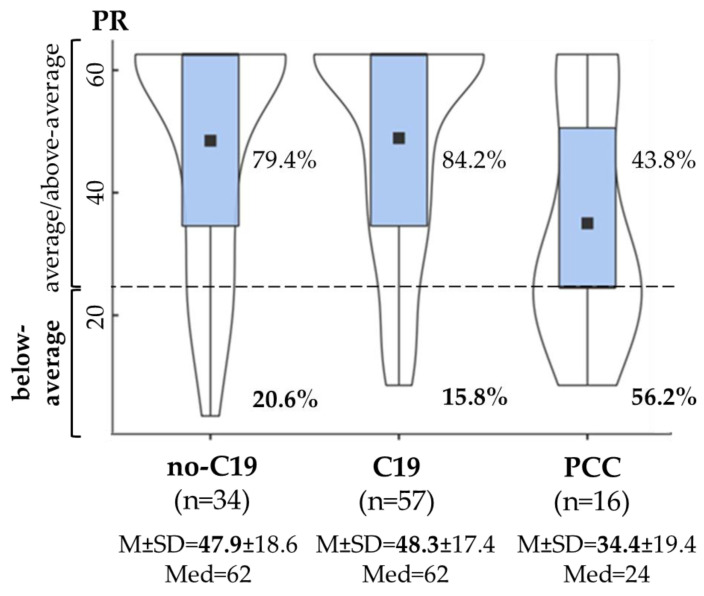
Verbal working memory in the no-C19 (COVID-19-negative controls), C19 (COVID-19-recovered without PCC) and PCC groups. Violin plots with density curves and boxplots of percentile rank (PR). The horizontal line in the boxplot is the median of the PR, the square is the mean of the PR. M ± SD: mean ± standard deviation, Med: median. Dashed line: PR = 25, below-average performance: PR < 25, average/above-average performance: PR ≥ 25.

**Figure 4 life-15-00821-f004:**
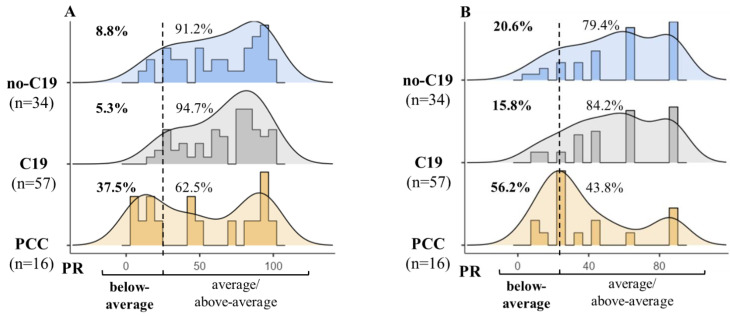
Frequency histograms of the subsidiary variables in verbal working memory in the no-C19 (COVID-19-negative controls), C19 (COVID-19-recovered without PCC) and PCC groups. Density curves of percentile rank (PR). Dashed line: PR = 25, below-average performance: PR < 25, average/above-average performance: PR ≥ 25. Subsidiary variables: (**A**): mean time “correct” (the mean reaction time for correct responses, “hits”) and (**B**): omitted (the number of omitted responses to a target stimulus, total missed stimuli).

**Figure 5 life-15-00821-f005:**
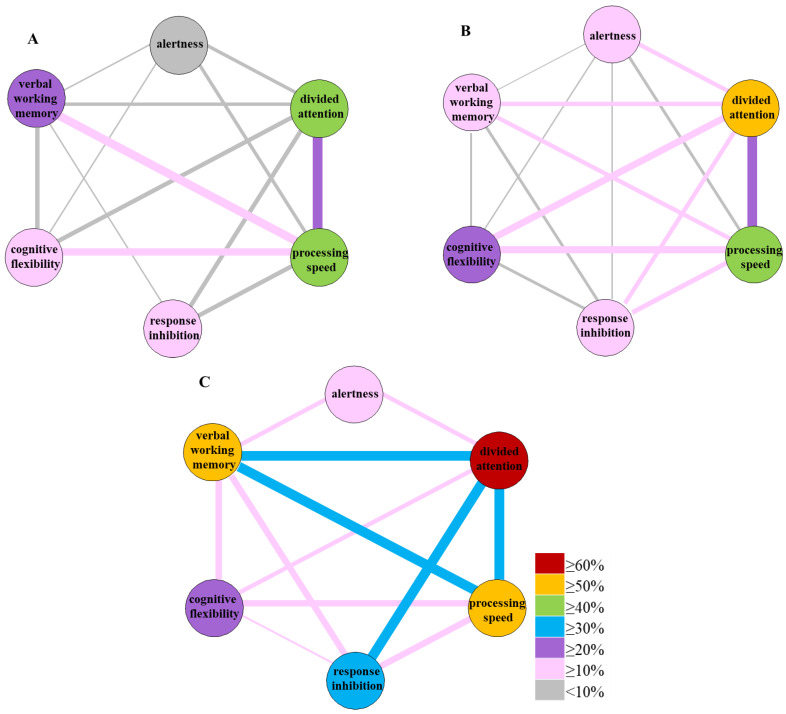
Co-occurrence of the decreased below-average performance of cognitive functions (%) in the no-C19 (COVID-19-negative controls; (**A**), n = 34), C19 (COVID-19-recovered without PCC; (**B**), n = 57) and PCC ((**C**), n = 16) groups. Line and circle color, as well as line thickness, correspond to the frequency indicated with different colors.

## Data Availability

The datasets generated and analyzed during the present study are available from the corresponding author.

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
