# Peer review of "Cognitive Slowing, Dysfunction in Verbal Working Memory, Divided Attention and Response Inhibition in Post COVID-19 Condition in Young Adults"

_life, 2025, doi:10.3390/life15050821_

Round 1

Reviewer 1 Report (Previous Reviewer 1)

Comments and Suggestions for Authors

I thank the authors for all the revisions and modifications made to improve the text.

Author Response

Reviewer 2 Report (Previous Reviewer 3)

Comments and Suggestions for Authors

The authors reviewed the manuscript in a consistence to the Reviewers indication.

The scientific work is logic and scientifically based.

Methods are appropriated.

The figures are clear.
Tables are informative.

The English language is fine.

Discussion consistent with results and limitations were well explained.

References appropriated.

Line 259: the notes to figure 4 should stay in one page

Line 278: figure 5: the figure should stay in one page with the footnotes

Author Response

Reviewer 3 Report (Previous Reviewer 4)

Comments and Suggestions for Authors

Major Comments:

  1. The operational specificity of the hypothesis on cognitive dysfunction lacks precision and is introduced too late (line 83). what domains? what tests?
  2. Despite excluding participants with documented comorbidities, the study fails to adequately account for psychosocial variables such as stress, anxiety, sleep quality, and academic workload—all of which are known to impact cognitive performance in young adults and could confound the observed associations.
  3. The sample size for the PCC group (n = 16) is notably small and undermines statistical power and generalizability. Although the authors used robust statistical tests, the effect sizes are small to moderate and may be driven by outliers or sample noise. A more balanced sample is required for credible between-group comparisons.
  4. The use of partial eta squared and Hedges’ g is acceptable, but with such small subgroups, these can seem deceptively large even in the presence of non-significant p-values.
  5. Without pre-infection cognitive data, it is speculative to attribute cognitive impairments directly to PCC. The authors rely solely on normative comparisons, which are insufficient to establish within-subject decline.
  6. Throughout the manuscript, the authors interpret non-significant trends (e.g., in divided attention and response inhibition) as indicative of cognitive deficits. This is misleading and undermines scientific rigor. Conclusions should be limited to statistically significant findings only.
  7. The study relies on retrospective self-report for symptom history, introducing a substantial risk of recall bias. Additionally, the recruitment of university students via convenience sampling introduces selection bias, yet these issues are under-addressed in the discussion.
  8. The classification of PCC is derived from self-reported WHO CRF criteria, without clinical verification. Given the heterogeneity of PCC, a more stringent and clinically validated diagnostic approach is needed.
  9. The authors begin with the claim that PCC-related deficits are objective (line 344), then concede that subjective reports do not always correlate (line 347), acknowledging some degree of inconsistency. More balance needs to be applied here.

Minor Comments:

  1. Figures such as 2–5 replicate information already conveyed in the text and tables. Their inclusion does not significantly enhance interpretation and may distract from the main findings.
  2. Terms such as "cognitive slowing" and "mental fatigue" are used loosely without clear operationalization. These need clearer definition and quantification within the study framework.
  3. The manuscript contains inconsistent citation formatting (e.g., “[10-12]” vs. “[20,25,30,32-34]”) and placeholder text (e.g., “To be added by editorial staff”) that should be addressed before publication.
  4. Beyond age and sex, key demographic factors such as socioeconomic status, academic field, and digital device use patterns—which may influence cognition—are omitted.

Author Response

This manuscript is a resubmission of an earlier submission. The following is a list of the peer review reports and author responses from that submission.

Round 1

Reviewer 1 Report

Comments and Suggestions for Authors

I would like to thank the authors for their work and the interest shown in a topic as relevant at the social and health care level as Post COVID-19 Condition (PCC) or persistent COVID.
After reading the study, I believe that there are several considerations that should be taken into account in order to improve its clarity/quality for publication.

- Firstly, it should be noted that the sample size calculation proposed =102 for the 3 groups has been made with a homogeneous distribution in each group of the number of participants, which is not consistent with the distribution of the study subjects between groups.
In the case of the present study, the distribution between groups is not homogeneous (non-C19 n=34, C19 -without PCC n=57 and PCC n=16), causing some groups to have twice or even three times as many participants as others. This poses a problem when comparing the data obtained, and could lead to an incovenience. This is why it seems to me to be an important limitation that should be included in the text.

-  Secondly, in Table S3 the statistically significant results are not marked or highlighted, which is advisable for the results tables.

Reviewer 2 Report

Comments and Suggestions for Authors

Dear Author,

I read and reviewed the present paper submitted to life.

Title of Manuscript: Cognitive Slowing, Dysfunction in Verbal Working Memory, Divided Attention and Response Inhibition in Post COVID-19 Condition in Young Adults

It is an interesting article; the authors investigate the cognitive abilities in post COVID-19 in youg adults (students). In fact, good idea, but some problems with the approach and the methods.

My comments:

  1. The introduction is well written, and the topic is well introduced.
  2. I doubt that the analysis using mean, SD and t-testing make sense in small groups with n=16 participants.
  3. No information on the background of the 16 post covid affected participants is reported. This can be the decisive information which could explain the observation. Familiar background, social background etc.
  4. Figure 1.: I do not understand why groups were selected like this 73 versus 16; the 16 are included in the 73, isn’t it?
  5. The violin plots are well prepared!
  6. A flow diagram would be helpful to understand the group selection.
  7. Based on the presented data, conclusions should be made with caution!!! It suggests post Covid as the single reason for the disturbance!
  8. What is the medical background of being contracted with SARS-CoV-2? Proved SARS-CoV-2 in one individual with or without infection? Or generally Infection?
Comments on the Quality of English Language

Dear Author,

I read and reviewed the present paper submitted to life.

Title of Manuscript: Cognitive Slowing, Dysfunction in Verbal Working Memory, Divided Attention and Response Inhibition in Post COVID-19 Condition in Young Adults

It is an interesting article; the authors investigate the cognitive abilities in post COVID-19 in youg adults (students). In fact, good idea, but some problems with the approach and the methods.

My comments:

  1. The introduction is well written, and the topic is well introduced.
  2. I doubt that the analysis using mean, SD and t-testing make sense in small groups with n=16 participants.
  3. No information on the background of the 16 post covid affected participants is reported. This can be the decisive information which could explain the observation. Familiar background, social background etc.
  4. Figure 1.: I do not understand why groups were selected like this 73 versus 16; the 16 are included in the 73, isn’t it?
  5. The violin plots are well prepared!
  6. A flow diagram would be helpful to understand the group selection.
  7. Based on the presented data, conclusions should be made with caution!!! It suggests post Covid as the single reason for the disturbance!
  8. What is the medical background of being contracted with SARS-CoV-2? Proved SARS-CoV-2 in one individual with or without infection? Or generally Infection?

Reviewer 3 Report

Comments and Suggestions for Authors

The manuscript life-3602919 entitled Cognitive Slowing, Dysfunction in Verbal Working Memory, Divided Attention and Response Inhibition in Post COVID-19 Condition in Young Adults by Johanna Takács, Darina Deák , Beáta Seregély , Akos Koller investigated the cognitive decline and function in Post Covid Conditions (PCC). A total of 107 young adults, university students, participated, aged 18–34 years. 68.2% had contracted SARS-CoV-2; 21.9% showed PCC. Three groups were compared: no-C19(non-COVID-19), C19(COVID-19), PCC. Attention and executive function were measured with Vienna Test System (Schuhfried®, Austria). In verbal working memory, the PCC group had a significantly lower performance with a moderate effect. The rate of below-average performance was higher in PCC(56.2%) compared to no-C19(20.6%) and C19(15.8%). In divided attention and response inhibition, PCC also showed lower performance, 62.5% and 37.5% respectively than no-C19 and C19. An impaired cognitive functions was more frequent in PCC. Verbal working memory was significantly impaired, and a lower performance was found in the divided attention and response inhibition. In addition, there was an increased reaction time in all cognitive tasks, demonstrating cognitive slowing in young people with PCC.

The manuscript is scientifically based.

Methods are appropriate

Tables and figures are clear

Results are god.

The English language is fine.

Discussion consistent with results and limitations were well explained.

References appropriated.

Line 136: the section title should stay with the following text

Figure 2: the violin plots are usually in vertical. The horizontal view is a bit strange.

Figure 3: the violin plots are usually in vertical. The horizontal view is a bit strange.

Line 224: figure 5 notes should stay with the figure in the previous page.

Reviewer 4 Report

Comments and Suggestions for Authors

  • The introduction provides a good overview, but the transition between PCC symptoms and the rationale for the study could be smoother. Explicitly state why studying cognitive function in young adults with PCC is important and how it fills a gap in the literature.
  • While the study carefully documents cognitive dysfunction in PCC, the manuscript lacks any mechanistic exploration. An in-depth discussion integrating current mechanistic hypotheses would be very valuable.
  • Describe testing conditions for cognitive assessments (e.g., noise, lighting, distractions) to ensure reproducibility.
  • Add references to support claims about prefrontal cortex involvement in PCC (e.g., neuroimaging or biomarker studies).
  • Acknowledge that the lack of pre-COVID cognitive data limits causal interpretations.
  • Clarify hypothesis: specify which cognitive domains, tests, and expected degree of dysfunction.
  • Time-since-infection (1.4 years ± 0.73) was not analyzed as a potential effect modifier; combining 6-month and 2-year cases is methodologically weak.
  • The sample is 79% female. This imbalance raises questions about the generalizability of findings. Sex differences in immune response are well-known. The authors should analyze and report whether sex affects the cognitive outcomes
  • No subgroup analysis based on time-since-infection was conducted.
  • No adjustment for multiple comparisons across cognitive domains; clarify or adjust statistical methods.
  • Report statistical test for the finding that 50.1% of PCC group showed below-average performance on ≥3 functions.
  • “Increased reaction time in all cognitive tasks”, please quantify this
  • Terms like “no-C19” and “C19” are somewhat informal. Standard terminology such as “COVID-negative controls” and “COVID-recovered without PCC” would improve clarity.
  • The authors say deficits “could not be explained” by fatigue or anxiety. However, this is speculative unless formally controlled for. Recommend rephrasing this claim or supporting with literature.
  • Grammar correction:
    • Abstract: “university students, participated, aged 18–34 years” → “university students aged 18–34 years participated.”
    • Page 8: “revealing below-average performance cognitive functions” → “revealing below-average performance in cognitive functions.”